# Powerful extragalactic jets dissipate their kinetic energy far from the central black hole

Adam Leah W. Harvey 🔵 [1✉], Markos Georganopoulos[1,2,3✉] & Eileen T. Meyer[1,3✉]

Accretion onto the supermassive black hole in some active galactic nuclei (AGN) drives relativistic jets of plasma, which dissipate a significant fraction of their kinetic energy into gamma-ray radiation. The location of energy dissipation in powerful extragalactic jets is currently unknown, with implications for particle acceleration, jet formation, jet collimation, and energy dissipation. Previous studies have been unable to constrain the location between possibilities ranging from the sub-parsec-scale broad-line region to the parsec-scale molecular torus, and beyond. Here we show using a simple diagnostic that the more distant molecular torus is the dominant location for powerful jets. This diagnostic, called the seed factor, is dependent only on observable quantities, and is unique to the seed photon population at the location of gamma-ray emission. Using 62 multiwavelength, quasi-simultaneous spectral energy distributions of gamma-ray quasars, we find a seed factor distribution which peaks at a value corresponding to the molecular torus, demonstrating that energy dissipation occurs ~1 parsec from the black hole (or ~$10^4$ Schwarzschild radii for a $10^9 M_\odot$ black hole).

[1] Department of Physics, University of Maryland, Baltimore County, 1000 Hilltop Circle, Baltimore, MD 21250, USA. [2] NASA Goddard Space Flight Center, 8800 Greenbelt Rd, Greenbelt, MD 20771, USA. [3]These authors jointly supervised this work: Markos Georganopoulos, Eileen T. Meyer. ✉email: aharvey1@umbc.edu; georgano@umbc.edu; meyer@umbc.edu

The spectral energy distribution (SED) of an extragalactic jet is well described by a general two-peak structure, with a low-energy component peaking in the infrared-optical due to synchrotron emission of the relativistic particles in the jet and a high-energy component peaking in the gamma-rays due to inverse Compton scattering by the same particles[1]. In powerful jets, the high-energy, gamma-ray component dominates over the low-energy component[2], and is the main outlet of energy dissipation. In powerful jets, the seed photons for inverse Compton scattering are thought to be external to the jet [e.g., refs. 3–5], and this mechanism is termed external Compton (EC) scattering. There are two primary populations of photons in powerful jet systems thought to be potential seeds for EC scattering: those from the sub-parsec-scale broad-line region, and those from the parsec-scale molecular torus [e.g., refs. 6,7].

Different methods of localizing the gamma-ray emission have been proposed, with contradictory results[1]. Short variability time-scales of gamma-ray emission in powerful jets has been used to argue that the dominant emission site must be on the sub-parsec-scale, implicating the broad-line region [e.g., ref. 8]. The broad-line region is opaque to very-high-energy (≥100 GeV) gamma-rays, and TeV detections of powerful jets have challenged the broad-line region scenario [e.g., refs. 9,10]. A lack of absorption features in the total average gamma-ray spectra of the most strongly *Fermi*-LAT detected powerful jets also challenges this scenario[11]. In contrast, several simultaneous gamma-ray/optical flares in which the optical emission exhibited polarization behavior similar to that of the very-long baseline interferometry (VLBI) radio emission implicates an emission site at or near the VLBI core, beyond the molecular torus [e.g., ref. 12]. Observations of both energy-dependent and energy-independent cooling times in gamma-ray flares of PKS 1510-089 implicate the molecular torus for some flares, and the VLBI core for other flares[13]. These prior results give contradictory answers for the site of gamma-ray emission, perhaps because they rely on single or few sources in a rare spectral state.

Here we develop and apply a diagnostic of the location of gamma-ray emission, which is dependent only on long-term average observable quantities, and which can be applied to a large sample of sources, avoiding many of the problems of prior methods. We call this diagnostic quantity the seed factor, and it is unique to the seed photon population at the location of emission. Because we cannot achieve short enough integration times at all wavebands to match the variability time of extremely fast variable states, we do not apply our seed factor diagnostic to fast variability. Fitting 62 quasi-simultaneous SEDs, we find that the molecular torus is strongly preferred as the dominant location of gamma-ray emission. This demonstrates that energy dissipation in powerful extragalactic jets occurs ~1 pc downstream of the supermassive black hole.

## Results

We have developed a diagnostic which we call the seed factor to test whether powerful extragalactic jets dissipate kinetic energy through EC in the broad-line region or the molecular torus. The derivation of the seed factor is given in the Supplementary Information. It is given by

$$\text{SF} = \text{Log10}\left(\frac{U_0^{1/2}}{\epsilon_0}\right) = \text{Log10}\left(3.22 \times 10^4 \frac{k_1^{1/2} \nu_{s,13}}{\nu_{c,22}} \text{ Gauss}\right) \tag{1}$$

where $U_0$ is the energy density of the photon population, which is upscattered by the jet, in CGS units; $\epsilon_0$ is the characteristic (i.e., peak) photon energy of the photon population, in units of the electron rest mass; $k_1$ is the Compton dominance (the ratio of the

inverse Compton peak luminosity to the synchrotron peak luminosity) in units of 10; $\nu_{s,13}$ is the peak frequency of the synchrotron peak in units of $10^{13}$ Hz; $\nu_{c,22}$ is the peak frequency of the inverse Compton component in units of $10^{22}$ Hz.

The seed factor value expected due to EC scattering on a specific photon population is calculated using the energy density and characteristic photon energy of the seed photon population to be upscattered. These are known very well for both the broad-line region and the molecular torus. Since seed factor values for the molecular torus and broad-line region differ significantly, the seed factor is unique to the photon population being upscattered. The seed factor of a specific source can be determined using four observable quantities from the broadband SED: the peak frequency and peak luminosity of the synchrotron and inverse Compton emission. These parameters are purely observationally derived and can be constrained well using existing data. For these reasons the seed factor diagnostic is very robust. After calculating the seed factor for a source, it can be compared to the expected value for either the broad-line region or molecular torus. Emission mechanisms other than external Compton scattering can also be tested against, since in such a case there is no a priori reason that the seed factor should be a single value for all sources or all emission states.

To calculate the seed factor expected for the broad-line region and the molecular torus (full details in Supplementary Information), an estimate of the energy density and characteristic photon energy of each photon population are needed. Reverberation mapping and near-infrared interferometric studies of radio-quiet AGN (e.g., refs. 14–17) combined with studies of the covering factors of the broad-line region[18] and molecular torus[19–21] imply that both the broad-line region and molecular torus have a constant energy density across sources. Combined with estimates of the characteristic photon energy of the broad-line region[22] and the molecular torus[23], the expected seed factor for the broad-line region and the molecular torus are, respectively,

$$\text{SF}_{\text{BLR}} = 3.29 \pm 0.11 \tag{2}$$

$$\text{SF}_{\text{MT}} = 3.92 \pm 0.11 \tag{3}$$

To investigate if there is a population trend in the location of energy dissipation, we study the distribution of the seed factor. The histogram of seed factors is plotted in Fig. 1. However, a histogram does not give any direct information about measurement uncertainties. To better visualize the measured seed factor distribution, we therefore implemented a kernel density estimation which incorporates the uncertainties via bootstrapping (see Sections 2.3 and 2.5 of the Supplementary Information). The kernel density estimate we have implemented uses Silverman's Rule[24] for the bandwidth to produce a reasonably unbiased smoothed kernel density estimate of the seed factor distribution (see Fig. 2). As can be seen from this kernel density estimate, the seed factor distribution peaks within the $1\sigma$ confidence interval of the molecular torus seed factor.

We further calculated the median of the seed factor distribution, and calculated the $1\sigma$ confidence interval for the median via bootstrapping (using $8 \times 10^5$ variates). The median seed factor is

$$\text{SF}_{\text{median}} = 4.01^{+0.10}_{-0.12} \tag{4}$$

We tested the seed factor distribution for normality and compared the distribution with that measured for weak extragalactic jets (which are known to emit through processes other than external Compton scattering) to test for consistency with external Compton scattering. Finding a rejection significance of normality of $1.42\sigma$ (using a bootstrapped two-sided Kolmogorov–Smirnov test) and finding that the weak and powerful jet distributions are statistically different, the seed factor

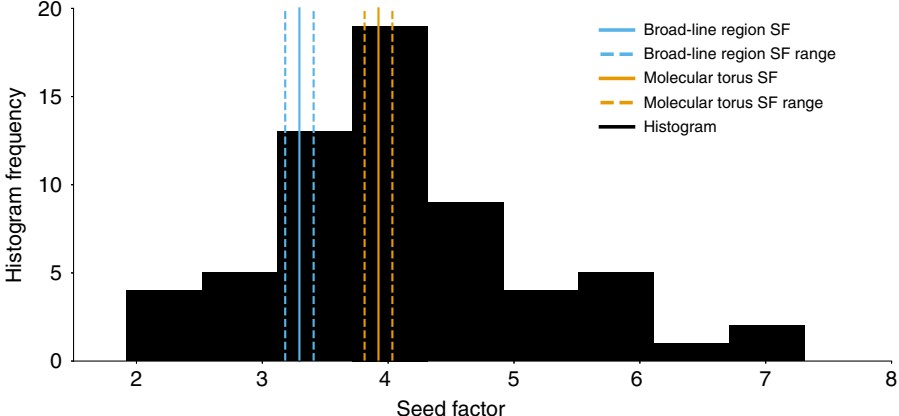

**Fig. 1 Histogram of FSRQ seed factors.** A histogram showing the distribution of seed factors for our sample of FSRQs. The histogram is plotted in black. The expected broad-line region seed factor and $1\sigma$ confidence interval are shown in blue, with solid and dashed lines, respectively, on the left. The expected molecular torus values are plotted similarly, on the right, in orange. See Sections 1.2 and 1.3 of the Supplementary Information for information on calculation of the plotted confidence intervals. The histogram was binned using the `auto` option for the binning in `matplotlib.pyplot.hist`. The histogram peaks at about the expected seed factor value of the molecular torus.

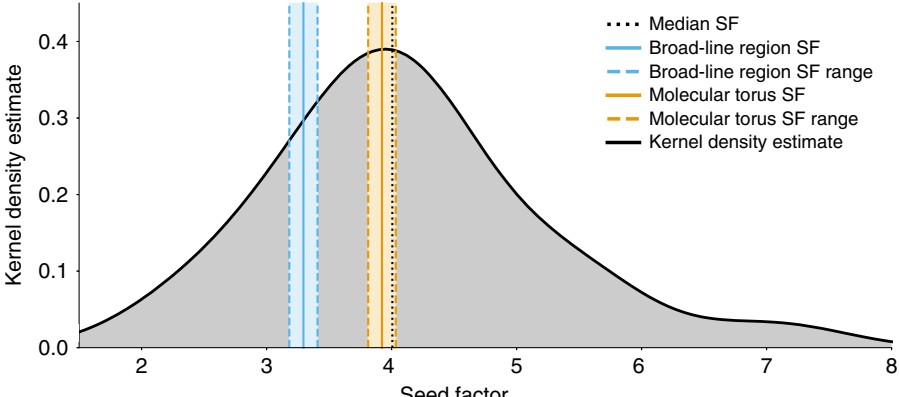

**Fig. 2 Kernel density estimate of FSRQ seed factors.** A kernel density estimate (KDE) of the seed factor distribution of the combined sample of 62 SEDs (see Sections 2.3 and 2.5 of the Supplementary Information for more details). The KDE is plotted as a thick black line, with the area underneath shaded in gray as a visual aid. The median of the distribution is denoted by a black dotted line. The expected broad-line region seed factor and $1\sigma$ confidence interval are shown in blue, with solid and dashed lines, respectively, on the left. The expected molecular torus values are plotted similarly, on the right, in orange. See Sections 1.2 and 1.3 of the Supplementary Information for information on calculation of the plotted confidence intervals. The KDE peaks within the $1\sigma$ confidence interval of the molecular torus, nearly coincident with the expected value of the molecular torus. The KDE has the general appearance of a normal distribution. A bootstrapped (using $10^7$ variates) two-sided Kolmogorov–Smirnov test indicates that normality cannot be rejected significantly (rejection significance of $1.42\sigma$; $p$ value 0.16).

distribution of powerful jets is consistent with external Compton scattering.

To test whether the seed factor distribution is inconsistent with emission from either the broad-line region or the molecular torus, we calculate the rejection significance of the expected seed factors. Using a bootstrapping method, we calculated the significance of rejecting that the median of the observed distribution is different from the respective expected values. The rejection significances calculated, in terms of standard deviations, for the broad-line region and the molecular torus, respectively, are,

$$\sigma(\mathrm{Obs_{median}} - \mathrm{BLR_{median}}) = 6.10\sigma \quad (5)$$

$$\sigma(\mathrm{Obs_{median}} - \mathrm{MT_{median}}) = 0.71\sigma \quad (6)$$

We therefore conclude that, except possibly in periods of fast variability, the molecular torus is strongly preferred as the single dominant location of energy dissipation in powerful extragalactic jets.

## Discussion

Our finding sets specific constraints on jet models: there is no substantial steady-state jet energy dissipation at scales less than ~1 pc. Within this distance the flow has to collimate and achieve an opening angle of a few degrees, and at the same time accelerate to bulk Lorentz factors of 10–50, as required by VLBI studies[25]. Major particle acceleration and subsequent dissipation of the order of 10% of the jet power[26] must take place beyond the sub-pc broad-line region and within the ~pc scale molecular torus. This conclusion does not rest on any single source, but rather on clear observables for an entire population of powerful jets. These observables are measured using quasi-simultaneous multiwavelength SEDs, which act as snapshots of the broadband emission of each source, reducing biases due to averaging. Quasi-simultaneous SEDs reduce the chance of interband integration mismatches. That is to say that a quasi-simultaneous SED is unlikely to, for example, contain X-ray data taken during a high state while other data is taken during a

low state. This work can be expanded as additional well-sampled SEDs become available.

We note here that the seed factor derived from a single SED of a single source would be reliable if we had near-perfect knowledge of the SED at all frequencies over exactly matching timeframes. This is essentially never the case. It is, however, possible to localize the emission for a single source with a large number of SEDs of that source.

A more detailed understanding of the broad-line region and the molecular torus would enable use of the seed factor to localize the site of energy dissipation more precisely, and to study the possible distribution of dissipation sites across both time and sources. The broad-line region and the molecular torus are expected both to be stratified [e.g., refs. [27,28]] and possibly exist as components of an accretion disk wind [e.g., ref. [29]]. Furthermore, minor deviations from the scaling relations we have used are currently under debate [e.g., refs. [14,17,30–32]]. The magnitude and nature of the effects of such a possible deviation would have implications for a more detailed analysis of the seed factor distribution. Conversely, the detailed structure of AGN hosting powerful jets can be constrained by requiring models to produce a seed factor distribution consistent with that observed.

Our result also argues that the VLBI core is not the dominant location of gamma-ray emission. The fact that the seed factor distribution has a peak and this peak is around the molecular torus seed factor would be a very unlikely coincidence if the emission location was typically the VLBI core.

## Methods

**Description of SED samples**. We used four samples (as described in Section 1.4 of the Supplementary Information) of well-sampled quasi-simultaneous, multi-wavelength SEDs of powerful blazars (flat spectrum radio quasars, FSRQS) to calculate the seed factor for a representative population of sources. We fit the published SEDs to obtain the peak frequencies and luminosities (these values are given in Supplementary Tables 2–6). To reliably constrain the peak luminosity and the peak frequency of the synchrotron and inverse Compton components, these SEDs needed to be well sampled in frequency (see Section 2.2 of the Supplementary Information for information on testing spectral coverage). Quasi-simultaneity is required due to the fact that blazars are variable sources, and we thus require the ability to reliably fit individual states of emission. Two of these samples are catalog samples from the literature, with SEDs published in their respective papers. One of these two samples is made up of five different subsamples selected based on different criteria in a project of the Planck Collaboration[33,34]. The second is the Fermi-LAT Bright AGN Sample[35]. The third is from a sample of blazar SEDs for some sources observed in the Tracking Active Galactic Nuclei with Austral Milliarcsecond Interferometry program (TANAMI[36]). The fourth sample is a sample of SEDs taken from a literature search of two different well-observed sources (3C 279 and 3C 454.3; see Section 1.4.4 of the Supplementary Information for more information). Some SEDs were excluded on the basis of poor coverage and scattering outside of the Thompson regime (that is, if $\nu_c > 10^{24}$ Hz; see Section 1.4 and 2.2 of the Supplementary Information for details on SED exclusions). In the case of SEDs from our literature search, SEDs were also excluded in the case of either non-quasi-simultaneous SEDs or observations of nonsteady states (i.e., when there was clear variability reported for the time of the observations; the seed factor is applicable only to steady-state emission).

**SED fitting**. We fit the obtained SEDs using maximum likelihood regression implemented through a simulated annealing optimization algorithm. Errors were calculated using the likelihood ratio method (i.e., Wilk's Theorem), implemented through a combination of bootstrapping, kernel density estimation, and polynomial interpolation (as described in the Supplementary Information). There are 62 SEDs in total, which were useable based on our criteria of quasi-simultaneity, coverage, steady-state emission, and Thomson regime scattering. We calculated the seed factor and 1σ uncertainties for these 62 SEDs.

The peak of any individual SED is only as reliable as the model and the data. The formal errors on the fits from application of Wilk's Theorem implicitly do not take into account the error from variability and the fact that the spectra are not intrinsically perfect polynomials but only well approximated as such. We assume that the extra error contribution averages out in the sample estimate of where the seed factor distribution peaks. This is the main reason that we do not attempt to make a claim about the location in any single source, and instead use many measurements to try to find the ensemble answer, which implicitly relies on the errors averaging out. This approach uses the population to provide statistical constraints on the behavior of powerful jets as a source class.

**Statistical analysis of the seed factor distribution**. We used a two-sided boot-strapped Kolmogorov–Smirnov test[37] to test the normality of the distribution, using a rejection significance threshold of 2σ. Normality of the seed factor distribution cannot be rejected significantly (rejection significance of 1.42σ; $p$ value of 0.16).

Consistency with external Compton scattering was tested by comparing the seed factor distribution of powerful extragalactic jets with that of weak extragalactic jets, which are known to emit through processes other than external Compton (see Section 4 of the Supplementary Information for more information). The distribution for powerful jets was found to be different than that for weak jets, further implying an external Compton origin of the emission from powerful jets. Due to the normality, sharp peak, and divergence from weak extragalactic jets, the observed seed factor distribution is consistent with a single, dominant location of energy dissipation in powerful extragalactic jets.

To test whether the seed factor distribution is inconsistent with either the broad-line region or the molecular torus, we calculated the significance of rejection of the respective expected seed factors. We implemented a bootstrapping procedure to produce a distribution of the difference between the median of the observed seed factor distribution and the expected seed factors. This was done by performing a large sample-size bootstrap on the seed factor distribution of the 62 SEDs, and creating a proxy sample of the same size for the broad-line region and molecular torus, drawing, for each, 62 random samples from their respective normal distributions (as defined by the calculated values and uncertainties given earlier). For each triplet of medians, we calculated the difference between the bootstrapped observed seed factor median and the proxy sampled broad-line region and molecular torus seed factor medians. Each distribution of differences is consistent with being normally distributed, as determined by a two-sided Kolmogorov–Smirnov test (using a rejection significance threshold of 2σ, normality is rejected at the 1.75σ ($p$ value 0.08) and 1.82σ ($p$ value 0.07) level for the broad-line region and molecular torus difference distributions, respectively). Given the normality of these distributions, the significance of rejecting the hypothesis that the medians are the same was then calculated as the absolute value of the ratio of the mean and standard deviation of each sample of differences (i.e., the number of standard deviations between zero and each median). The rejection significances of the seed factor differences, in terms of standard deviations, for the broad-line region and molecular torus, respectively, are,

$$\sigma(\mathrm{Obs_{median}} - \mathrm{BLR_{median}}) = 6.10\sigma \tag{7}$$

$$\sigma(\mathrm{Obs_{median}} - \mathrm{MT_{median}}) = 0.71\sigma \tag{8}$$

**Reporting summary**. Further information on research design is available in the Nature Research Reporting Summary linked to this article.

## Data availability

All data used are publicly available. Data available as machine-readable tables can be found using the following DOI names: LBAS: 10.26093/cds/vizier.17160030 [https://doi.org/10.26093/cds/vizier.17160030]; Giommi: 10.26093/cds/vizier.35410160 [https://doi.org/10.26093/cds/vizier.35410160]; DSSB: 10.26093/cds/vizier.35910130 [https://doi.org/10.26093/cds/vizier.35910130]. Some data were not available as machine-readable tables, but were extracted from plots (using Dexter[38]) or tables in published papers, as noted in the Supplementary Information. The digitized tables are available upon request.

## Code availability

Results can be reproduced using standard free analysis packages. Methods are fully described. Code can be made available by request. Version and accession information for publicly available software packages referenced are as follows: matplotlib.pyplot.hist (matplotlib 2.2.3) [https://matplotlib.org/2.2.3/api/_as_gen/matplotlib.pyplot.hist.html#matplotlib.pyplot.hist]; scipy.interpolate.splrep (scipy 1.2.1) [https://docs.scipy.org/doc/scipy/reference/generated/scipy.interpolate.splrep.html#scipy.interpolate.splrep]; scipy.interpolate.sproot (scipy 1.2.1) [https://docs.scipy.org/doc/scipy/reference/generated/scipy.interpolate.sproot.html#scipy.interpolate.sproot]; sklearn.neighbors.KernelDensity (scikit-learn 0.19.2) [https://scikit-learn.org/0.19/modules/generated/sklearn.neighbors.KernelDensity.html#sklearn.neighbors.KernelDensity]; gammapy (gammapy 0.9) [https://docs.gammapy.org/0.9/]; Dexter (Dexter 0.5a) [http://dexter.sourceforge.net/]; Simulated Annealing (Simulated Annealing 3.2) [https://eml.berkeley.edu/Software/abstracts/goffe895.html].

Published online:

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

## Acknowledgements

A.L.W.H. and M.G. acknowledge support from *Fermi* grant NNX14AQ71G.

## Author contributions

A.L.W.H. collected all data from the literature and developed and performed all relevant analyses. M.G. and E.T.M. supervised the work.

## Competing interests
The authors declare no competing interests.
