## [Peer Review File · Nature Communications]

Reviewers' comments:

Reviewer #1 (Remarks to the Author):

Referee Report on Nature Communications Manuscript NCOMMS-20-00647-T

"Powerful extragalactic jets dissipate their kinetic energy far from
the central black hole"

by A. L. W. Harvey, M. Georganopoulos, and E. T. Meyer

=====

In this manuscript, the authors suggest a diagnostic of the dominant seed photon field for external Compton scattering in powerful blazars (FSRQs), based purely on the observed peak frequencies and fluxes of the synchrotron and inverse-Compton components in the SEDs of these blazars, in the paradigm of a leptonic jet model. The diagnostic quantity, the Seed Factor, is expected to assume characteristic values for the BLR and the Dusty Torus (DT) as target photon fields, and the authors find that the distribution of Seed Factors derived from the observed SEDs is statistically significantly consistent with the seed photons coming from the DT and inconsistent with the BLR hypothesis. They draw the conclusion that this indicates a location of the gamma-ray emission region in powerful blazars at pc-scale distances from the central black hole, outside the BLR.

On a very general note: This is an interesting new idea to diagnose the seed photon fields for EC scattering in blazars which will deserve publication after a couple of comments and questions listed below have been addressed. However, there have been many earlier indications that the DT photon field at least plays a significant role for SED modelling of FSRQs, and the lack of BLR gamma-gamma absorption features also suggests a location of the gamma-ray emission region near the outer boundary or beyond the BLR. As I further detail below, it is also not clear whether a significant contribution of the BLR photon field may not simply be suppressed by Klein-Nishina effects so that EC on the dust torus dominates the gamma-ray SED. Therefore, while the study definitely confirms that the DT photon field plays an important role in shaping the SEDs of FSRQs, this result may not be a unique diagnostic on the actual location of the gamma-ray emission region. As such, I am not convinced that the manuscript is of sufficiently broad interest to readers outside the field of astronomy/astrophysics to warrant publication in Nature Communications. Instead, I see this as a very good paper for a specialized astrophysics journal. If the authors wish to stick with the Nature outlets, maybe Nature Astronomy would be an appropriate choice.

Specific comments / questions:

1) I am concerned about the Thomson-scattering assumption: First of all, Klein-Nishina effects really already set in at $\epsilon_0 \gamma \sim 1/4$ (rather than 1), where ϵ_0 is the seed photon energy in the emission-region rest frame. With that, Klein-Nishina effects become important already at $\nu_c \sim 1e23 \text{ Hz} / (1 + z)$ for scattering of BLR photons. Many values of ν_c in tables 1 - 6 actually violate this limit, indicating that KN effects could have been important in a scenario with BLR-dominated EC. This would lead the authors' method to under-estimate, in particular, ϵ_0 , thus, over-estimate the SF. Therefore, a BLR-dominated EC scenario might simply be mis-interpreted as DT-dominated by the authors' diagnostic. Alternatively, KN effects could also just suppress the BLR-EC spectral signatures so that they wouldn't dominate the gamma-ray SED. Yet, the emission region could still be located in the region of influence of the BLR photon field.

2) Following up on 1), it also needs to be pointed out that the redshift dependence has been neglected in Eqs. (9) - (11). I understand that it cancels out in the final evaluation of the SF, but for considerations of the Thomson vs. Klein-Nishina limit, it is important.

3) Further on the values in tables 1 - 6: I find it a reason for concern that there are objects with multiple SEDs, for which the inferred SFs vary over a large range. For example, for 3C279, they range from 2.8 to 4.5; for 0537-441, they range from 2.36 to 4.83. Now, assuming that ϵ_0 does not change dramatically (if it is always DT photons), this would imply that U_0 changes by ~ 4 orders of magnitude between

different states! What is the authors' interpretation / explanation of this?

4) It would also be nice to see the complete set of SED plots and fits in the supplemental material. I am wondering, in particular, about SEDs where the authors find EC peaks at < 100 MeV, where no data exist. How confidently can the data actually constrain EC peaks in that energy range? I am a bit skeptical about the small error bars given on those values - and seeing the SED plots might alleviate my concerns.

In summary: While I think this is an interesting new idea for a diagnostic

of EC scattering in blazars, I believe that there are still some caveats that render the interpretation in terms of location of the gamma-ray emission region less convincing than the authors seem to believe. I therefore suggest to (a) submit this paper to a specialized astronomy/astrophysics journal and (b) mention the caveats listed above in the manuscript.

Reviewer #2 (Remarks to the Author):

Major Concerns:

1. Authors have implemented a simple concept to understand an important aspect of blazar jet emission physics. However, they need to justify the significance and novelty of their work better in order to show the relevance and applicability of their work in a broader context. They have failed to assert how this diagnostic will influence thinking in the field, how is it a novel claim when previous studies have alluded to molecular torus and beyond to being a plausible location of gamma-ray emission in such powerful blazars (see e.g., Marscher et al. (2010), Dotson et al. (2015), and given that the seed factor diagnostic is applicable only to steady-state emission of powerful blazars, how will it be applicable to a more generalized scenario of variable emission?
2. Authors have failed to discuss how their approach will deal with cases where a correlation between gamma-ray and radio emission has been observed. How will the seed factor distribution take such cases into account?
3. Authors need to state that the seed factor diagnostic is applicable only to steady-state emission early on, maybe in the Introduction itself instead of mentioning it within parentheses in Section 4. This is an important aspect of the applicability of their diagnostic and must be discussed in the beginning.

4. The authors must also define steady-state. In addition, they must clarify that if their study targets only steady-state emission then how is the issue of the location of gamma-ray emission even relevant here because steady-state emission does not exhibit short-term variability on time scales of hours to minutes in the GeV regime?

5. Lines 48-49: The statement that the “models which may be degenerate in their physical parameters” needs more explanation. Are the authors trying to suggest that it is the degeneracy in model parameters that might be resulting in contradictory answers for the site of gamma-ray emission?

6. Line 51: Authors have used long-term average of observable quantities to develop their diagnostic. However, average gets skewed by any extreme values that might be present in the data set. How have authors taken care of that?

7. Lines 56-57: The result that “energy dissipation in powerful extragalactic jets occurs ~ 1 pc downstream” seems biased because the VLBI core as a plausible location for gamma-ray emission has not been considered in this study and neither has been justified that why has that not been considered.

8. At various places, authors have used rejection significance of normality but nowhere have they stated the threshold level that they are using above which the result will be considered significant. For clarity, that value needs to be mentioned. In other words, what is the chosen alpha value for such cases below which the null hypothesis is rejected?

9. Lines 134-136: It is not clear from those statements how will using quasi-simultaneous SEDs reduce biases due to averaging?

10. Line 151: How is “well-sampled” defined in their context? For the sake of completeness, it would be good to provide sample SEDs from each of the samples that have been considered for this study.

11. Line 168: Since this study is only valid for Thomson regime and is a limitation of this diagnostic, its implications must be discussed briefly in Section 3.

12. Lines 196-215: The description of the implementation of bootstrapping and calculation of rejection significances, in its current format, is quite confusing. It needs to be rewritten to properly understand

a. How the three distributions were constructed? The way I have understood it currently, it seems to me that - Distribution 1 (Dist1) = distribution of observed seed factors Distribution 2 (Dist2) = distribution of expected seed factors Distribution 3 (Dist3) = Dist1 – Dist2

b. Is the mean and standard deviation being calculated for only Dist3 or for 1 & 2 as well?

c. What are (Median_Dist1 – Median_Dist2) & (Median_Dist1 – Median_Dist3)

being used for?

d. Is Mean of Dist3/Standard Deviation of Dist3 > 3 sigma being checked in this

process?

e. In Eq. (7) & (8) what are the values of Obs_median, BLR_median, & MT_median?

f. With 62 samples, shouldn't the Student t-distribution be more appropriate than Gaussian? This needs to be discussed further.

g. Are the number of data points (62 in this case) featuring in the final sigma test?

For example, $\sqrt{62} * \text{mean} / \text{standard deviation}$?

13. Bootstrapping also needs to be discussed briefly in terms of

a. How many times the process of sampling with replacement was repeated?

b. What were the assumptions that were made while undertaking bootstrap analysis (such as independence of samples)?

c. What was the bootstrap scheme that was used for this analysis (it seems Case Resampling) but it needs to be stated in the text.

d. Whether they calculated the standard error of the mean resulting from the sampling distribution or simply the standard deviation?

14. Line 313: The steady-statedness needs to be defined in this context because the SEDs of 3C 279 & 3C 454.3, as given in Table 5 of the manuscript, include periods of flares as well.

15. Fig. 3: Do the lines correspond to the expected values of seed factor for FSRQs? If yes, it needs to be stated clearly in the figure caption. If no, then the expected values of seed factor for BL Lacs look quite similar to that of FSRQs and that needs to be justified in the text.

16. It should be mentioned explicitly that the fitted SEDs were used to obtain the values of peak frequency and luminosity of synchrotron and Compton bumps in Tables 1 – 5, otherwise it's easy to confuse those values with the ones given in the references, e.g. 35, where these SED samples were constructed from.

Minor Concerns:

1. The language needs to be improved further.

2. There are a few typos throughout the manuscript.
3. The URLs do not work properly in the Reference section.
4. In Abstract, line 13, please add "seed" before "photon population" to distinguish it from the photon population internal to the jet.
5. Line 52: according to the authors the method is applicable to a large sample of sources. It would be good if the authors can briefly discuss the applicability of their diagnostic to a single source and what might be the limitation(s) of the method they might encounter in such a scenario.
6. Lines 73-34: Need references for both broad line region and molecular torus for those cases.
7. Lines 80-84: So if the value doesn't match either of the expected values then that means that emission is not due to external Compton scattering, correct? In that case the seed factor value would not match the corresponding values of seed factor for either the broad line region or the molecular torus. In such a scenario, how would we know the location of the gamma-ray emission? This point must be discussed briefly in the text.
8. At few places, such as lines 144, 317, citations should be given in ascending order.
9. For completeness, in Eq. (13), please provide expressions for L_c and L_s .
10. For completeness, please provide expressions for energy density U_0 & characteristic photon energy ϵ_0 .

Reviewer #3 (Remarks to the Author):

This manuscript presents a fresh argument for constraining the distance scale of energy dissipation and production of blazar emission in the powerful relativistic jets launched by the active galactic nuclei (AGN). A new parameter called the seed factor (SF) is introduced and estimated for a sample of 62 "gamma-ray quasars", also known as the flat spectrum radio quasars (FSRQs). The SF values determined observationally from the broad-band spectral energy distributions (SED) of these FSRQs can be compared with the reference values predicted theoretically by the external radiation Comptonization (ERC) model for two major components of AGN radiation fields produced externally

with respect to the jet that dominate at distinct distance scales, namely the broad emission lines (BEL) (produced in the broad line region (BLR)) at distances of ~ 0.1 pc, and thermal IR radiation (produced in the molecular tori (MT)) at distances of ~ 1 pc. The Authors conclude that the observed SF distribution favours the MT distance scales, being statistically inconsistent with the BLR scales.

This work has a potential to become significant in the decades-long discussion on the origin of blazar emission. I find the basic argument to be valid, while the main conclusion is not exactly as definite as the Authors suggest (see Comment 1 below). Considering other issues listed below, my recommendation is that these results are definitely worth publishing, but not necessarily in a Nature journal.

Major comments:

1. I would argue that time-averaged SEDs may indeed be produced predominantly at the MT distance scales, however, rapid flares of blazar emission, that are routinely observed in those sources, require higher values of external radiation energy density (U_0) that cannot be provided by IR radiation, but can well be provided by the BEL at the BLR scales (e.g., ApJ, 789, 161). In fact, the Authors admit in Section 4 (end of paragraph 1) that:

"(...) SEDs were also excluded in the case of either non-quasi-simultaneous SEDs or observations of non-steady states (i.e., when there was clear variability reported for the time of the observations; the seed factor is applicable only to steady-state emission)."

I would also argue that there is no contradiction between steady-state blazar emission produced at the MT scales and rapid blazar flares produced at the BLR scales. However, this new result should not be understood as if dissipation in the FSRQ jets can *only* occur at the MT scales. In this sense I would strongly disagree with this sentence from Section 3, paragraph 1:

"Major particle acceleration and subsequent dissipation of the order of 10% of the jet power must take place beyond the sub-pc broad-line region and within the \sim pc scale molecular torus."

2. Although the theoretical SF values are very robust, the distribution of observationally determined SF values is extremely broad, spanning 5 orders of magnitude (Figure 1). This should raise a question on the basic underlying assumption that all these SEDs can be effectively produced in single and uniform emitting regions. What is the nature of those outlier SEDs that suggest $SF \sim 2$ or $SF \sim (5-7)$, can they be produced in a single emitting region? Is it really possible to say that such a broad observed distribution is consistent with a very well defined SF value for the ERC of the MT radiation? Another potentially complicating factor, the synchrotron self-absorption, is not mentioned at all.

3. I would also argue that while the concept of the SF parameter is novel, all necessary ingredients for its derivation can be found in the publication ApJ, 704, 38. In particular, the key reason for the robustness of the SF is that the observed frequency ratio and the luminosity ratio (the Compton dominance) for the two main SED components in the FSRQs can be related without degeneracy from the magnetic field strength or the jet Lorentz/Doppler factor, has been shown in their Eq. (52). I have an impression that the robustness of the SF parameter is not emphasized clearly enough in the manuscript.

Minor comments:

4. In Section 5.1, it is not mentioned in which reference frame are the parameters measured (the word "frame" is not found anywhere). It appears that B is the magnetic field in the jet co-moving frame. Indicating this would also be helpful in Section 2, following Eq. (1), in the case of U_0 and ϵ_0 .

5. In Eqs. (9-10), the γ symbol is not explained.

6. References should be provided in the last sentence of the first paragraph of Section 1, at least to the two seminal works on Comptonization of BLR and MT photons: ApJ, 421, 153 and ApJ, 545, 107.

Nature Communications Review Response

We thank the referees for the time they took reviewing our work and for their insightful and constructive reports. Below we respond to their reports point by point and mention the lines where we modified the manuscript to address their considerations. To keep further communication clear, we have numbered the comments/questions requiring a response.

1 First Referee Report

(1) Referee 1: *In this manuscript, the authors suggest a diagnostic of the dominant seed photon field for external Compton scattering in powerful blazars (FSRQs), based purely on the observed peak frequencies and fluxes of the synchrotron and inverse-Compton components in the SEDs of these blazars, in the paradigm of a leptonic jet model. The diagnostic quantity, the Seed Factor, is expected to assume characteristic values for the BLR and the Dusty Torus (DT) as target photon fields, and the authors find that the distribution of Seed Factors derived from the observed SEDs is statistically significantly consistent with the seed photons coming from the DT and inconsistent with the BLR hypothesis. They draw the conclusion that this indicates a location of the gamma-ray emission region in powerful blazars at pc-scale distances from the central black hole, outside the BLR.*

On a very general note: This is an interesting new idea to diagnose the seed photon fields for EC scattering in blazars which will deserve publication after a couple of comments and questions listed below have been addressed. However, there have been many earlier indications that the DT photon field at least plays a significant role for SED modeling of FSRQs, and the lack of BLR gamma-gamma absorption features also suggests a location of the gamma-ray emission region near the outer boundary or beyond the BLR.

Response: We agree with the referee that this work fits in with/confirms in a novel way something which has been suggested in previous works. Our work stands out in being population-based, rather than based on one or a few sources, and in the robustness of the seed factor. Previous works argue either on exclusion (that the variable gamma-ray emission is *not* produced in the sub-pc BLR) or, based on one source, that it is produced in the VLBI core at a distance of a few to several pc from the supermassive black hole (SMBH).

We do specifically mention/cite the lack of BLR gamma-ray absorption in the introduction currently (paragraph 2, Costamante et al. 2018). This important work shows that the emission site is beyond the BLR but it does not address the question of if the emission site is within the pc-scale DT or at the several pc environment where the VLBI core is located. In contrast, our work shows explicitly for the first time that the gamma-ray emission of a collection of powerful blazars is preferred to originate within the DT.

(2) Referee 1: *As I further detail below, it is also not clear whether a significant contribution of the BLR photon field may not simply be suppressed by Klein-Nishina effects so that EC on the dust torus dominates the gamma-ray SED. Therefore, while the study definitely confirms that the DT photon field plays an important role in shaping the SEDs of FSRQs, this result may not be a unique diagnostic on the actual location of the gamma-ray emission region. As such, I am not convinced that the manuscript is of sufficiently broad interest to readers outside the field of astronomy/astrophysics to warrant publication in Nature Communications. Instead, I see this as a very good paper for a specialized astrophysics journal. If the authors wish to stick with the Nature outlets, maybe Nature Astronomy would be an appropriate choice.*

Response: We agree that this is an important point – see our full response to the question of the Klein-Nishina effects below in response to comment 3. Briefly, we show that the effect of the KN cross-section does not significantly affect our FSRQ sample, because the observables it is based on are formed in the Thomson regime. This is verified for individual sources in the sample.

(3) Referee 1: *I am concerned about the Thomson-scattering assumption: First of all, Klein-Nishina effects really already set in at $\epsilon'_0\gamma \approx 1/4$ (rather than 1), where $\epsilon'_0 \approx \Gamma\epsilon_0$ is the seed photon energy in the emission-region rest frame. With that, Klein-Nishina effects become important already at $\nu_c \approx 10^{23}/(1+z)$ Hz for scattering of BLR photons. Many values of ν_c in tables 1 - 6 actually violate this limit, indicating that KN effects could have been important in a scenario with BLR-dominated EC. This would lead the authors' method to under-estimate, in particular, ϵ_0 , thus, over-estimate the SF. Therefore, a BLR-dominated EC scenario might simply be mis-interpreted as DT-dominated by the authors' diagnostic. Alternatively, KN effects could also just suppress the BLR-EC spectral signatures so that they wouldn't dominate the gamma-ray SED. Yet, the emission region could still be located in the region of influence of the BLR photon field.*

Response: We thank the referee for bringing up this issue. It also assisted us in finding an error: previously in the entire manuscript we adopted $\epsilon_{BLR} = 3 \times$

10^{-5} (following Tavecchio & Ghisellini 2008, MNRAS 386, 945), except for the calculation of the energy above which scattering takes place in the KN regime, where we used $\epsilon_{BLR} = 10^{-4}$. We have corrected this to use the first value. Taking into account the point of the referee for the lower-energy onset of KN effects, and including the redshift factor, we obtain a frequency $3.5 \times 10^{23}/(1+z)$ Hz as the maximum energy that a Thomson treatment is justified. Using this, we found only one of the FSRQ SEDs to have $\nu_c > 3.5 \times 10^{23}/(1+z)$ Hz. This is 1424-418 with $\log \nu_c = 23.18$ from Table 1, about 10% above the limit. As our FSRQ sources are in the Thomson regime for both seed photon fields, we are safe in not considering KN effects in this work. We have added a clarification of this issue to the manuscript in section 5.1., lines 234-239.

(4) Referee 1: *It also needs to be pointed out that the redshift dependence has been neglected in Eqs. (9) - (11). I understand that it cancels out in the final evaluation of the SF, but for considerations of the Thomson vs. Klein-Nishina limit, it is important.*

Response: We have added the redshift dependence in section 5.1, equations 9-11.

(5) Referee 1: *Further on the values in tables 1 - 6: I find it a reason for concern that there are objects with multiple SEDs, for which the inferred SFs vary over a large range. For example, for 3C279, they range from 2.8 to 4.5; for 0537-441, they range from 2.36 to 4.83. Now, assuming that ϵ_0 does not change dramatically (if it is always DT photons), this would imply that U_0 changes by ~ 4 orders of magnitude between different states! What is the authors' interpretation / explanation of this?* **Response:** The values of ϵ_0 and U_0 for

the BLR and the MT are fixed. The seed factors that correspond to the BLR and the MT are therefore fixed. Spread in the measured seed factors is expected due to uncertainties in our four observables. An order of magnitude total uncertainty in any observable quantity is probably a fair conservative estimate. This is why we use many measurements to try to find the “wisdom of crowds” answer, which implicitly relies on the errors averaging out.

(6) Referee 1: *It would also be nice to see the complete set of SED plots and fits in the supplemental material. I am wondering, in particular, about SEDs where the authors find EC peaks at < 100 MeV, where no data exist. How confidently can the data actually constrain EC peaks in that energy range? I*

am a bit skeptical about the small error bars given on those values - and seeing the SED plots might alleviate my concerns.

Response: We appreciate the suggestion for clarity on this point, and have now included a complete set of SED plots in the supplementary material to the paper. The formal errors on the fits come from an application of Wilk's Theorem, and do not take into account the (very difficult to quantify) error from variability and the fact that the spectra are not intrinsically perfect polynomials but only well-approximated by that shape. It is an implicit assumption here that the extra error contribution averages out in the sample estimate of where the seed factor peaks. This is the main reason that we do not attempt to make a claim about the location in any single source.

Although no data exists near 100 MeV, the peak can still be constrained near this energy assuming there is X-ray and GeV data. The shape of the SED is constrained by the data that exists. If there is enough data to reliably fit the curvature (our criterion for energy coverage), then we presumably have enough data to reliably fit the peak frequency and peak flux. We also assume that the fit of the SEDs is never perfect. So the peak of any individual SED is only as reliable as the model and the data. However, this work relies on a population of SEDs for precisely this reason. We do not take the values to be completely accurate for any individual source. We instead rely on the errors averaging out, and use the population for statistical constraints.

2 Second Referee Report

(1) Referee 2: Authors have implemented a simple concept to understand an important aspect of blazar jet emission physics. However, they need to justify the significance and novelty of their work better in order to show the relevance and applicability of their work in a broader context. They have failed to assert how this diagnostic will influence thinking in the field, how is it a novel claim when previous studies have alluded to molecular torus and beyond to being a plausible location of gamma-ray emission in such powerful blazars (see e.g., Marscher et al. (2010), Dotson et al. (2015), and given that the seed factor diagnostic is applicable only to steady-state emission of powerful blazars, how will it be applicable to a more generalized scenario of variable emission?

Response: Most previous work has focused on variable emission and flares in a single source. We focus instead on the more common states of emission in a population context. Some previous work (e.g., Costamante et al. 2018 MNRAS 477 4749) that has been done on studying a population of blazars is able to exclude the BLR, but cannot discriminate between the MT and further out at

the VLBI core distances. Contrary to that, our approach, that is dependent on only observable quantities gives a positive result for the MT.

To clarify an important point: the main issue regarding applicability to variable versus "steady" emission states is that we need the time interval during which data is collected to be shorter than the variability time so that the observables derived from the SEDs are consistent and accurate. Because we cannot achieve short enough integration times at all wavebands to match the variability time of extremely fast variable states, we do not apply our seed factor diagnostic to fast variability. We have amended the introduction to reflect this point.

(2) Referee 2: *Authors have failed to discuss how their approach will deal with cases where a correlation between gamma-ray and radio emission has been observed. How will the seed factor distribution take such cases into account?*

Response: In external Compton models, correlation between the radio and gamma-rays are expected due to beaming effects of orientation, and indeed the form of the correlation for powerful blazars suggests external Compton (Meyer et al., 2012). For the seed factor, the radio is not really relevant, since it is not one of the observables (in blazar models, the radio is generally regarded as coming from different (larger) scales than the higher-energy synchrotron emission around the peak). Regarding correlated variability between radio and gamma-ray, we do not suggest that our approach is applicable to fast variability.

(3) Referee 2: *Authors need to state that the seed factor diagnostic is applicable only to steady-state emission early on, maybe in the Introduction itself instead of mentioning it within parentheses in Section 4. This is an important aspect of the applicability of their diagnostic and must be discussed in the beginning.*

Response: We have altered the introduction to explain better the applicability of the diagnostic (as noted in response to question 1, above).

(4) Referee 2: *The authors must also define steady-state. In addition, they must clarify that if their study targets only steady-state emission then how is the issue of the location of gamma-ray emission even relevant here because steady-state emission does not exhibit short-term variability on time timescales of hours to minutes in the GeV regime.*

Response: In the blazar community, the terms steady-state and flare are both rather loosely used. In the case of our study, it is not that our diagnostic is

not applicable to variable states, but that accuracy requires a consistent set of observations, representative of the same spectral state, at all wavebands. This is technically difficult during very fast flares.

(5) **Referee 2:** *Lines 48-49: The statement that the “models which may be degenerate in their physical parameters” needs more explanation. Are the authors trying to suggest that it is the degeneracy in model parameters that might be resulting in contradictory answers for the site of gamma-ray emission?*

Response: What we mean is that in a self-consistent one-zone model there are more model parameters than observational constraints, so different sets of model parameters can give very similar SEDs. In this sense, it is possible that a given SED can be modeled with different seed photons.

Also, in many cases one-zone modeling is done by selecting by hand an electron energy distribution with an arbitrary break energy and slope change at the breaks (not solving the electron kinetic equation). This gives the modeler additional freedom, at the price of a non self-consistent physical description.

However as this is not the main point of the paper we have removed the statement.

(6) **Referee 2:** *Line 51: Authors have used long-term average of observable quantities to develop their diagnostic. However, average gets skewed by any extreme values that might be present in the data set. How have authors taken care of that?*

Response: The LBAS has a γ -ray integration time of 88 days for all SEDs. A companion paper (Abdo et al. 2010 ApJ 722 520) investigated the weekly lightcurves of the LBAS sources (note that 2 of the sources in our paper are missing in their data) in a 47 week period starting at the start of the LBAS period. We have calculated the normalized excess variance ($F_{var} = \sqrt{(S^2 - \langle \sigma_F^2 \rangle) / \langle F \rangle}$) for the 13 weeks overlapping with the LBAS period, the normalized excess variance for the total 47 weeks, and compared these two quantities by calculating the fractional difference between them. This is detailed in the table below. Of note are (1) the normalized excess variance for all sources is below 1, with a maximum of about 0.8, and a median of about 0.4 (2) the fractional difference is only negative (i.e., the LBAS period has a larger excess variance than the total 47 weeks) for one source, and the median of the fractional difference is about 0.3, implying that the excess variance is generally a sizeable fraction smaller during the LBAS period than over the entire 47 week period (i.e., there is less variability in the integration time of the LBAS than these sources experience

Table 1: LBAS Normalized Excess Variances

OFGL Name	LBAS Time EV	Total EV	Fractional Difference of EV
J0457.1-2325	0.23	0.48	0.53
J1512.7-0905	0.80	0.95	0.15
J1256.1-0547	0.15	0.78	0.81
J2143.2+1741	0.37	0.51	0.27
J0349.8-2102	0.27	0.71	0.61
J1457.6-3538	0.58	0.60	0.04
J1229.1+0202	0.51	0.61	0.17
J2254.0+1609	0.35	0.92	0.62
J1522.2+3143	0.26	0.30	0.16
J0531.0+1331	0.43	0.81	0.47
J1504.4+1030	0.63	0.40	-0.57

on average).

The DSSB SEDs were created via a Bayesian block method based on Fermi-LAT lightcurves to select periods of relatively steady flux.

The other samples were integrated over similarly short lengths of time, and thus it is likely that variability is similarly small and/or averaged out.

(7) Referee 2: *Lines 56-57: The result that “the energy dissipation in powerful extragalactic jets occurs ~ 1 pc downstream” seems biased because the VLBI core as a plausible location for gamma-ray emission has not been considered in this study and neither has been justified that why has that not been considered.*

Response: We agree that the VLBI core is a viable candidate for the location of energy dissipation in some cases of fast variability, but our result argues that it is not the dominant location based on our population results. The fact that the SF has a peak and this peak is around the MT seed factor would be a very unlikely coincidence if the emission location was typically the VLBI core. We have added a comment along these lines in the end of Section 3 of the manuscript.

(8) Referee 2: *At various places, authors have used rejection significance of normality but nowhere have they stated the threshold level that they are using above which the result will be considered significant. For clarity, that value needs to be mentioned. In other words, what is the chosen alpha value for such cases below which the null hypothesis is rejected.*

Response: We thank the reviewer for bringing this to our attention. This value should have been quoted, and was taken as 2σ , which equates to $\alpha = 0.05$. This has been added to the manuscript. The same rejection significance threshold was used for the Anderson-Darling and Shapiro-Wilk tests in Section 5.3. This has been updated there as well.

(9) Referee 2: *Lines 134-136: It is not clear from those statements how will using quasi-simultaneous SEDs reduce biases due to averaging?*

Response: We simply meant that quasi-simultaneous SEDs reduce the chance of inter-band integration mismatches. That is to say that a quasi-simultaneous SED is unlikely to, for example, have x-ray data during a high state while other data is taken during a low state.

(10) Referee 2: *Line 151: How is "well-sampled" defined in their context? For the sake of completeness, it would be good to provide sample SEDs from each of the samples that have been considered for this study.*

Response: Well-sampled in the context of our paper was defined using spectral curvature. SEDs were excluded if a power-law was preferred over a log-parabola at the $\geq 2\sigma$ level. We chose to use curvature as a proxy for "well-sampledness" since blazar spectral components are expected to have curvature. Therefore, if an SED cannot significantly constrain the curvature of either the synchrotron or IC component, the SED must not be well-sampled (either due to actual wavelength coverage or due to measurement uncertainties). A statistical test for what could be called "effective coverage" is preferred over observer choice, since it reduces the possibility of any observer bias and places all SEDs on an equal statistical footing.

Postage-stamp plots of all of the SEDs used have been added. The figures are contained in the relevant section of the manuscript.

(11) Referee 2: *Line 168: Since this study is only valid for Thomson regime and is a limitation of this diagnostic, its implications must be discussed briefly in Section 3.*

Response: Please see our reply to reviewer 1, comment 2.

(12) Referee 2: Lines 196-215: The description of the implementation of bootstrapping and calculation of rejections significances, in its current format, is quite confusing. It needs to be re-written to properly understand.

a. How the three distributions were constructed? The way I have understood it currently, it seems to be that- Distribution1 (Dist1) = distribution of observed seed factors Distribution 2 (Dist2) = distribution of expected seed factors Distribution 3 (Dist3) = Dist1 - Dist2

b. Is the mean and standard deviation being calculated for only Dist3 or for 1 & 2 as well?

c. What are (Median_Dist1 - Median_Dist2) & (Median_Dist1 - Median_Dist3) being used for?

d. Is Mean of Dist3/Standard Deviation of Dist3 > 3 sigma being checked in this process?

e. In Eq. (7) & (8) what are the values of Obs_median, BLR_median, & MT_median?

f. With 62 samples, shouldn't the Student t-distribution be more appropriate than Gaussian? This needs to be discussed further.

g. Are the number of data points (62 in this case) featuring in the final sigma test? For example, $\sqrt{62} * \text{mean} / \text{standard deviation}$?

Response: We have added sections on bootstrapping and error analysis to manuscript (6.2 Bootstrapping Methods; 6.3 Error Analysis), which we believe address all of the above points.

(13) Referee: Bootstrapping also needs to be discussed briefly in terms of

a. How many times the process of sampling with replacement was repeated?

b. What were the assumptions that we made while undertaking bootstrapping?

c. What was the bootstrap scheme that was used for this analysis (it seems Case Resampling) but it needs to be stated in the text.

d. Whether they calculated the standard error of the mean resulting from the sampling distribution or simply the standard deviation?

Response:

a. This is addressed by sections 6.2 Bootstrapping Methods and 6.3 Error Analysis which were added to the manuscript.

b. The major assumption on the data is that the measurements are independent. Other assumptions can be found in Babu & Rao (The Indian Journal of Statistics 2004, 66, 1, 63-74) (these are generally just reasonable assumptions, such as uncertainty being asymptotically linear).

c. Case resampling (known by a variety of other names) is essentially what we have done, and is the standard bootstrap used in literature. However, the process is modified to incorporate uncertainties. These details have been added to the manuscript. Furthermore, the bootstrapping for the KS tests is different, and uses a procedure as defined in Feigelson and Babu (2012) (as cited in the manuscript).

d. This is described in the text. The uncertainty for the SF median is the standard error (of the median, not the mean).

(14) Referee 2: *Line 313: The steady-statedness needs to be defined in this context because the SEDs of 3C 279 & 3C 454.3, as given in Table 5 of the manuscript, include periods of flares as well.*

Response: Please see our previous reply regarding our removal of previously published states of fast variability.

(15) Referee 2: *Fig. 3: Do the lines correspond to the expected values of seed factor for FSRQs? If yes, it needs to be stated clearly in the figure caption. If no, then the expected values of seed factor for BL Lacs look quite similar to that of FSRQs and that needs to be justified in the text.*

Response: The lines correspond to the seed factor expected in FSRQs. Since BL Lacs do not have a BLR or MT, we did not attempt to predict the distribution of values for BL Lacs of the function of observables corresponding to the seed factor. We have clarified that it is for FSRQs in the caption (note that Figure 3 is now Figure 8). We have also updated the wording in the caption of Figure 1 to ensure that it is clear that the orange line represents the expected MT seed factor range (we had simply said the molecular torus is plotted similarly).

(16) Referee 2: *It should be mentioned explicitly that the fitted SEDs were used to obtain the values of peak frequency and luminosity of synchrotron and Compton bumps in Tables 1-5, otherwise it's easy to confuse those values with the ones given in the references, e.g. 35, where these SED samples were constructed from.*

Response: We have updated the text to make this explicit (see bold text in

lines 56 and 163 – 164 of the manuscript).

(17) Referee 2: *In Abstract, line 13, please add “seed” before “photon population” to distinguish it from the photon population internal to the jet.*

Response: This has been updated as requested.

(19) Referee 2: *Lines 73-34: Need references for both broad line region and molecular torus for those cases.*

Response: Extensive references for the values used in calculating these quantities is given in lines 89-93, which we think is the more appropriate location. Citing in lines 73-74 would require repeating 10 citations, which do not directly state the claim we here make. The fact that these quantities are known very well is given by the fact that they can be calculated from just the well-constrained results of the relevant references.

(20) Referee 2: *At few places, such as lines 144, 317, citations should be given in ascending order.*

Response: Citations are given in ascending order throughout the text. The citations in the lines listed were used earlier in the manuscript. Thus they have lower numbers than other citations in nearby, preceding lines.

(21) Referee 2: *Line 52: according to the authors the method is applicable to a large sample of sources. It would be good if the authors can briefly discuss the applicability of their diagnostic to a single source and what might be the limitation(s) of the method they might encounter in such a scenario.*

Response: We focus on a population due to issues explained in comment 5 (referee 1). The seed factor derived from an SED of a single source would be reliable if we had near-perfect knowledge of the SED at all frequencies over exactly matching timeframes. This is essentially never the case. It is, however, possible to localise the emission with a large number of SEDs for a given source. We added this comment in lines 143-148 of the manuscript.

(22) Referee 2: *Lines 80-84: So if the value doesn't match either of the expected values then that means that emission is not due to external Compton scattering, correct? In that case the seed factor value would not match the corresponding values of seed factor for either the broad line region or the molecular torus. In such a scenario, how would we know the location of the gamma-ray emission? This point must be discussed briefly in the text.*

Response: Our work presents a diagnostic and its application to constraining the location of gamma-ray emission. If the seed factor distribution were not consistent with the values expected for either the BLR or MT, it would imply that emission occurs outside both. However it would not provide information about the location beyond this. Our work could not constrain the location in such a case.

As we found that the seed factor is in fact consistent with the molecular torus (and not the BLR), discussion of how to locate the site of emission in a hypothetical case which would disagree directly with our observations is beyond the scope of this work.

Note: There were some additional minor comments regarding definitions or grammar which we have addressed.

3 Third Referee Report

(1) Referee 3: *I would argue that time-averaged SEDs may indeed be produced predominantly at the MT distance scales, however, rapid flares of blazar emission, that are routinely observed in those sources, require higher values of external radiation energy density (U_0) that cannot be provided by IR radiation, but can well be provided by the BEL at the BLR scales (e.g., ApJ, 789, 161). In fact, the Authors admit in Section 4 (end of paragraph 1) that: "(...) SEDs were also excluded in the case of either non-quasi-simultaneous SEDs or observations of non-steady states (i.e., when there was clear variability reported for the time of the observations; the seed factor is applicable only to steady-state emission)."*

*I would also argue that there is no contradiction between steady-state blazar emission produced at the MT scales and rapid blazar flares produced at the BLR scales. However, this new result should not be understood as if dissipation in the FSRQ jets can *only* occur at the MT scales. In this sense I would strongly disagree with this sentence from Section 3, paragraph 1: "Major particle accel-*

eration and subsequent dissipation of the order of 10% of the jet power must take place beyond the sub-pc broad-line region and within the pc scale molecular torus.”

Response: We agree with the referee that the matter of the emission location during less common fast-variability states remains unsettled, and that this point should be made clear in our conclusion. We have edited the concluding sentence in the main body of the paper to address this point.

(2) Referee 3: *Although the theoretical SF values are very robust, the distribution of observationally determined SF values is extremely broad, spanning 5 orders of magnitude (Figure 1). This should raise a question on the basic underlying assumption that all these SEDs can be effectively produced in single and uniform emitting regions. What is the nature of those outlier SEDs that suggest $SF \sim 2$ or $SF \sim (5 - 7)$, can they be produced in a single emitting region? Is it really possible to say that such a broad observed distribution is consistent with a very well defined SF value for the ERC of the MT radiation? Another potentially complicating factor, the synchrotron self-absorption, is not mentioned at all.*

Response: Due to measurement uncertainties, the seed factor of any individual SED does not provide strong constraints on the underlying nature of the observed state. We draw implications only from the population as a whole. This has been expounded upon in further detail in response to question 3 of Referee 1. Regarding SSA, it is only relevant at radio frequencies that do not affect the synchrotron peak frequency or luminosity.

(3) Referee 3: *I would also argue that while the concept of the SF parameter is novel, all necessary ingredients for its derivation can be found in the publication ApJ, 704, 38. In particular, the key reason for the robustness of the SF is that the observed frequency ratio and the luminosity ratio (the Compton dominance) for the two main SED components in the FSRQs can be related without degeneracy from the magnetic field strength or the jet Lorentz/Doppler factor, has been shown in their Eq. (52). I have an impression that the robustness of the SF parameter is not emphasized clearly enough in the manuscript.*

Response: We thank the referee for discussing this paper of Sikora et al 2009. The SF is derived from elementary theoretical considerations, so it is no surprise that similar equations have been found by others, such as in the paper you mentioned and which we now cite in Section 5.1. Our contribution is the realization

that a particular form of these equations can be used to localize the emission site by using only observables. We now include a comment on the robustness of the SF parameter in lines 81-83 of the manuscript.

(4) Referee 3: *In Section 5.1, it is not mentioned in which reference frame are the parameters measured (the word "frame" is not found anywhere). It appears that B is the magnetic field in the jet co-moving frame. Indicating this would also be helpful in Section 2, following Eq. (1), in the case of U_0 and ϵ_0 .*

Response: This has been addressed in the manuscript.

(5) Referee 3: *In Eqs. (9-10), the γ symbol is not explained.*

Response: This has been addressed in the manuscript.

(6) Referee 3: References should be provided in the last sentence of the first paragraph of Section 1, at least to the two seminal works on Comptonization of BLR and MT photons: ApJ, 421, 153 and ApJ, 545, 107.

Response: We have added these citations, thank you for the suggestion.

REVIEWER COMMENTS

Reviewer #1 (Remarks to the Author):

The authors have addressed all my points to my satisfaction. I am therefore happy to recommend the manuscript for acceptance.

Reviewer #2 (Remarks to the Author):

I thank the authors for clarifying most of the issues associated with the previous version. However, there are still a few areas that need further clarification and must be addressed before the manuscript is ready for publication.

1. I agree with Referee 1 regarding the suitability of this manuscript in Nature Communications. I'm still not convinced that it is suitable for this journal. However, Nature Astronomy, if not a specialized astrophysics journal, seems to be a much better choice for this work to get published.

2. It would be good if authors can include their nicely written response to #6 about ensuring that variability doesn't affect their analysis, somewhere in the main text, maybe Section 4.

3. Response to #9 on quasi-simultaneous SEDs is a good point and should be included in the text at an appropriate location.

4. Thanks for clarifying the meaning of well-sampled in the context of their work in response to #10. This should also be included at an appropriate location in the text.

5. Expressions for $L_c, L_s, U_0, \epsilon_0$ have not been provided. I understand that they are well known quantities, but for the sake of building an easy reference of equations for the derivation of the seed factor it would be good to include their corresponding expressions here as well.

Reviewer #3 (Remarks to the Author):

I would like to thank the Authors for improving the manuscript and for addressing most of my previous comments. It is particularly helpful that all the individual SEDs are now included, as requested by other Reviewers.

I can now answer my previous question about the outliers. For example, consider the "Giommi FSRQ" sample (Table 3, Figure 3). The source J0136+4751 has a very high SF ~ 7.3 , and the source J1924-2914 has a very low SF ~ 1.9 . We can see that this is driven mainly by very different IC frequency peaks: $\log_{10}(\nu_{IC}) \sim 18.0$ vs. 23.6 . Figure 3 shows that these two sources have in fact very similar observed SEDs, with one key difference: J1924-2914 has been detected in \sim GeV gamma-rays by the Fermi/LAT, and J0136+4751 has not (possibly due to higher redshift). We can see that the synchrotron components are very similar, and even the X-ray data are very similar with noticeable curvature. The very narrow IC component model for J0136+4751 appears to be driven by the X-ray curvature, while the very wide IC component model for J1924-2914 completely ignores the X-ray curvature. This example demonstrates how unreliable can individual SF estimates be. I would like to request that such a discussion point should be included somewhere in the manuscript.

This manuscript is valuable for introducing a formal definition of the seed factor, however, the significance of this result is limited by the quality of the observational datasets used. I agree with Reviewer 1 that it should rather appear in Nature Astronomy.

Two additional minor requests:

1. In Figures 3-7 (the SEDs) please indicate the estimated SF value on every panel. That would greatly help to understand the outliers.

2. In Figure 7, there is no reason to have horizontal error bars. The observational frequencies are known precisely.

Nature Communications Review Response 2

We thank the referees again for their time and constructive responses.

1 Second Referee Report

(1) Referee 2: *I agree with Referee 1 regarding the suitability of this manuscript in Nature Communications. I'm still not convinced that it is suitable for this journal. However, Nature Astronomy, if not a specialized astrophysics journal, seems to be a much better choice for this work to get published.*

Response: We thank the referee for their opinion.

(2) Referee 2: *It would be good if authors can include their nicely written response to 6 about ensuring that variability doesn't affect their analysis, somewhere in the main text, maybe Section 4.*

Response: The reference response has been added to the manuscript in the Supplemental Information as Section 5.3 Impact of Variability in SED Samples.

(3) Referee 2: *Response to 9 on quasi-simultaneous SEDs is a good point and should be included in the text at an appropriate location.*

Response: The response has been added to the manuscript in Section 3 within the lines noted in the original response.

(4) Referee 2: *Thanks for clarifying the meaning of well-sampled in the context of their work in response to 10. This should also be included at an appropriate location in the text.*

Response: The response has been added to the manuscript in the Supplemental Methods as Section 6.3 Effective Coverage of SEDs. A note has also been

made in the Methods section directing readers to this new section for information on determining "sampledness" of an SED. We also added a reference to Supplemental Methods near the same location where we had before erroneously only referenced Supplemental Information.

(5) Referee 2: *Expressions for L_c , L_s , U_0 , ϵ_0 have not been provided. I understand that they are well known quantities, but for the sake of building an easy reference of equations for the derivation of the seed factor it would be good to include their corresponding expressions here as well.*

Response: We give now the expressions for L_c and L_s just before equation (13). U_0 and ϵ_0 are characteristics of the broad line region and, in the context of this work take either the BLR or the MT values, as we discuss in the supplement.

2 Third Referee Report

(1) Referee 3: *I can now answer my previous question about the outliers. For example, consider the "Giommi FSRQ" sample (Table 3, Figure 3). The source J0136+4751 has a very high $SF = 7.3$, and the source J1924-2914 has a very low $SF = 1.9$. We can see that this is driven mainly by very different IC frequency peaks: $\log_{10}(nu_{IC}) = 18.0$ vs. 23.6 . Figure 3 shows that these two sources have in fact very similar observed SEDs, with one key difference: J1924-2914 has been detected in GeV gamma-rays by the Fermi/LAT, and J0136+4751 has not (possibly due to higher redshift). We can see that the synchrotron components are very similar, and even the X-ray data are very similar with noticable curvature. The very narrow IC component model for J0136+4751 appears to be driven by the X-ray curvature, while the very wide IC component model for J1924-2914 completely ignores the X-ray curvature. This example demonstrates how unreliable can individual SF estimates be. I would like to request that such a discussion point should be included somewhere in the manuscript.*

Response: A comment regarding this point has been added to the manuscript in the Methods section at lines 206-217.

(2) Referee 3: *This manuscript is valuable for introducing a formal definition of the seed factor, however, the significance of this result is limited by the quality*

of the observational datasets used. I agree with Reviewer 1 that it should rather appear in Nature Astronomy.

Response: We thank the referee for their opinion.

(3) Referee 3: *In Figures 3-7 (the SEDs) please indicate the estimated SF value on every panel. That would greatly help to understand the outliers.*

Response: We thank the referee for this suggestion, which should greatly improve the utility of the SED plots for readers. We have done as suggested in the updated manuscript.

(4) Referee 3: *In Figure 7, there is no reason to have horizontal error bars. The observational frequencies are known precisely.*

Response: As is common in the high-energy astrophysics community, horizontal error bars here are used to represent the binning and spectral resolution induced ranges on the observations. We therefore prefer to keep these error bars in the plots to provide a general understanding of the effective frequency bins of the SEDs (where such information has been supplied in the relevant samples). A comment regarding this has been added to the beginning of Section 5.2 Description of SED Samples.

REVIEWER COMMENTS

Reviewer #2 (Remarks to the Author):

I thank the authors for addressing all my concerns. I now recommend the publication of this manuscript.

Reviewer #3 (Remarks to the Author):

I insist that horizontal error bars as large as a decade in frequency in certain panels of Figure 7 make absolutely no sense, especially in the radio-to-optical bands, but also in the X rays, and even in the gamma rays they are too large. For some reason, they only appear for the source 3C 279, please check again publication 2012ApJ...754..114H, where most of these SEDs are taken from.

Nature Communications Review Response 3

We thank all the referees again for their contribution to helpful input.

1 Third Referee Report

(1) Referee 3: *I insist that horizontal error bars as large as a decade in frequency in certain panels of Figure 7 make absolutely no sense, especially in the radio-to-optical bands, but also in the X rays, and even in the gamma rays they are too large. For some reason, they only appear for the source 3C 279, please check again publication 2012ApJ...754..114H, where most of these SEDs are taken from.*

Response: We thank the referee for pointing out this issue. We have omitted the horizontal error bars (and the corresponding note in Section 5.2) as per the referee's request. Upon investigation, it seems that the frequency bands for the data from that particular paper were indeed displayed incorrectly. While this display error did impact the presentation of the band energy range for some SEDs in the corresponding supplemental figures, this information was not used in any way in our actual analysis, and we agree that the horizontal error bars could be confusing and are not necessary.

REVIEWERS' COMMENTS

Reviewer #3 (Remarks to the Author):

Thank You. I can now recommend publication of this manuscript. I did suggest publishing this in Nature Astronomy, but this is up to the Editors.

Nature Communications Review Response 4

We thank all of the referees for the time, effort, and thought they gave to reviewing our work.

1 Third Referee Report

(1) Referee 3: *Thank You. I can now recommend publication of this manuscript. I did suggest publishing this in Nature Astronomy, but this is up to the Editors.*

Response: We thank the referee for their opinion.